# BiMLP: Compact Binary Architectures for Vision Multi-Layer Perceptrons

**Yixing Xu, Xinghao Chen, Yunhe Wang**
Huawei Noah's Ark Lab
{yixing.xu, xinghao.chen, yunhe.wang}@huawei.com

## Abstract

This paper studies the problem of designing compact binary architectures for vision multi-layer perceptrons (MLPs). We provide extensive analysis on the difficulty of binarizing vision MLPs and find that previous binarization methods perform poorly due to limited capacity of binary MLPs. In contrast with the traditional CNNs that utilizing convolutional operations with large kernel size, fully-connected (FC) layers in MLPs can be treated as convolutional layers with kernel size $1 \times 1$. Thus, the representation ability of the FC layers will be limited when being binarized, and places restrictions on the capability of spatial mixing and channel mixing on the intermediate features. To this end, we propose to improve the performance of binary MLP (BiMLP) model by enriching the representation ability of binary FC layers. We design a novel binary block that contains multiple branches to merge a series of outputs from the same stage, and also a universal shortcut connection that encourages the information flow from the previous stage. The downsampling layers are also carefully designed to reduce the computational complexity while maintaining the classification performance. Experimental results on benchmark dataset ImageNet-1k demonstrate the effectiveness of the proposed BiMLP models, which achieve state-of-the-art accuracy compared to prior binary CNNs. The MindSpore code is available at `https://gitee.com/mindspore/models/tree/master/research/cv/BiMLP`.

## 1 Introduction

Recent years have witness the boosting of convolutional neural networks (CNNs) on several computer vision (CV) applications, *e.g.*, image recognition [19, 38, 14, 9, 11], object detection [35, 34], semantic segmentation [32] and low-level vision [21]. However, the success of CNN models highly depends on their huge computational cost and massive parameters, which are not suitable to be directly applied to edge devices that have limited computational resources such as mobile phones, smart watches.

There are several model compression and acceleration methods to reduce the number of parameters and FLOPs of the original CNNs and derive a portable model. For instance, knowledge distillation [16, 48, 49] aims to train a small student network with the help of a cumbersome teacher network. Filter pruning [29, 15, 42, 41] methods sort the weights of the network based on their importance and throw away those who have negligible influence on the final performance. Model quantization [4, 54, 31] methods reduce the original 32-bit floating point weights and activations into lower bits and tensor decomposition [1, 46] methods express a large weight tensor as a sequence of elementary operations on a series of simpler tensors. Among them, binary CNN [53, 33, 12, 45] is an extreme case of model quantization that uses only 1-bit for weights and activations. Compared to the original convolutional operation, binarized one has $64\times$ less FLOPs and $32\times$ less memory consumption.

36th Conference on Neural Information Processing Systems (NeurIPS 2022).

Note that all of the existing binarization methods are applied on the convolutional operations in CNN models. However, multi-layer perceptron (MLP) model also shows its potential on CV tasks [43, 3, 10] and matrix multiplication is computational friendly to GPUs and has advantage on inference time in reality. Compared to traditional CNNs that utilize convolutional operation with various kernel sizes, FC layers in MLP models can be treated as convolutions with kernel size $1 \times 1$, which suffer more from limited representation ability compared to the convolutional operations with larger kernel size when being binarized. In fact, directly binarizing MLP models with the existing methods show more severe performance degradation compared to the CNN models. For example, the Top-1 classification accuracy on ImageNet will drop by more than 23% when binarizing CycleMLP [3] and Wave-MLP [40] with the method proposed in Dorefa-Net [53] while the performance drop is only 17% for ResNet-18 [14] using $3 \times 3$ kernel size for most of the convolutional layers and 13% for AlexNet [19] with larger kernel size ($11 \times 11$ and $5 \times 5$).

Thus, in order to alleviate the problem mentioned above, we introduce a novel binary block for MLP model that contains multiple branches to merge a series of outputs from the same stage of the network. Besides, a universal shortcut connection (Uni-shortcut) is designed to better transfer the knowledge from previous layers to the current layer. Compared to the original shortcut connection, the proposed Uni-shortcut can merge two features with different shapes. By fusing the outputs of multiple layers derived from the same or different stages of the network, we show that the representation power of the binary MLP is drastically increased. The downsampling layers are also modified to reduce the computational complexity while maintaining the classification performance. Ex-

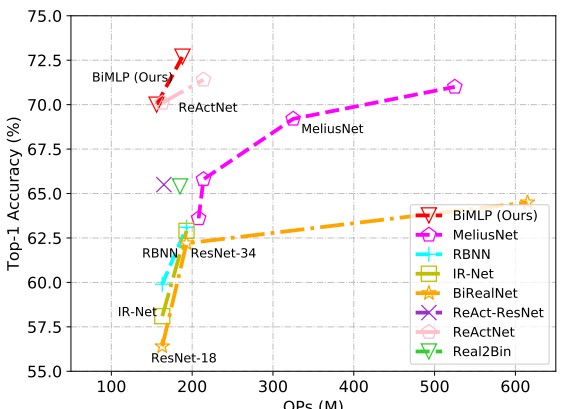

Figure 1: Accuracy vs. OPs of different binary models on ImageNet-1k dataset.

perimental results on ImageNet-1k dataset demonstrate that by using the proposed method, we achieve state-of-the-art performance compared to other binary CNN models, as shown in Fig. 1.

We summarize our contributions for learning 1-bit vision MLPs as follows:

- This paper points out the main difficulty of binarizing MLP models is that the representation ability of FC layers is worse than the convolutional layers in CNN models with kernel size larger than $1 \times 1$.

- We introduce a mulit-branch binary MLP block (MBB block) with Uni-shortcut operation to unlock the representation power of binary MLP models and also modify the architecture of downsampling layer to reduce the computational complexity.

- The experimental results on ImageNet-1k dataset show that the proposed BiMLP improves the top-1 accuracy by 1.3% with 12.1% less OPs compared to the state-of-the-art ReActNet, which indicates the effectiveness of the proposed BiMLP models.

## 2  Preliminaries

### 2.1  Binarization in CNN models

Given the weights $W \in \mathbb{R}^{c_o \times c_i \times k \times k}$ where $c_i$ and $c_o$ are the number of input and output channels and $k$ is the kernel size, and also the activation $A \in \mathbb{R}^{c_i \times h \times w}$ where $h$ and $w$ are the height and width of the input feature, a common way of binarizing the weights and activations in CNN models is to apply sign function on the 32-bit floating point inputs, $i.e.$,

$$W_b = sign(W), \quad A_b = sign(A), \tag{1}$$

in which $sign(\cdot)$ is an element-wise operation that outputs +1 when the input is positive and -1 otherwise. Given the binarized weights and activations, the convolutional operation can be realized

with binary operations only, *i.e.*:

$$Y = W_b \odot A_b, \tag{2}$$

in which $\odot$ represents binary convolutional operation that can be implemented with XNOR and POPCOUNT.

Note that the gradient of aforementioned sign function is zero almost everywhere and backward propagation is unable to be applied during training. Thus, the back-propagation of sign function follows the straight through estimator (STE) rule:

$$\frac{\partial \mathcal{L}}{\partial W} = Clip(\frac{\partial \mathcal{L}}{\partial W_b}, -1, 1), \tag{3}$$

where $\mathcal{L}$ is the loss function corresponding to the specific task, *e.g.*, cross-entropy loss for image classification. The $Clip(\cdot, -1, 1)$ function clips the elements of the gradient into range $[-1, 1]$.

Most of the existing binary CNN methods focus on improving the forward sign function and backward STE rule. For example, XNOR-Net [33] introduces channel-wise scale factors on the pure sign function and improves the performance of binary CNN. Dorefa-Net [53] uses only a single scale factor to achieve similar performance. ReActNet [26] proposes RSign and RPreLU function for binary CNNs. DSQ [8] replaces the STE rule and uses the gradient of tanh function to replace the gradient of sign function while BNN+ [6] introduces SignSwish function. Methods mentioned above are all applied on CNN models and directly transfer them to MLP models results in severe performance degradation.

## 2.2 Vision MLP Models

Vision MLP models take patches of image (tokens) as input, and stack a sequence of channel-FC layers and spatial-FC layers for image recognition. Specifically, given the input $X \in \mathbb{R}^{n \times d}$ in which $n$ is the number of tokens and $d$ is the number of channels in each token, the channel-FC layer fuses the channels in a single token and generate $d'$ channels, *i.e.*,

$$\text{channel-FC}(X, W_c) = X \cdot W_c, \tag{4}$$

where $W_c \in \mathbb{R}^{d \times d'}$ is the parameter matrix of the channel-FC layer, and $\cdot$ represents the matrix multiplication operation. Channel-FC layer only aggregates information from different input channels, while lacking the communications between tokens. Thus, the spatial-FC layer is also introduced in MLP models [43], *i.e.*,

$$\text{spatial-FC}(X, W_s) = W_s^\top \cdot X, \tag{5}$$

where $W_s \in \mathbb{R}^{n \times n'}$ is the parameter of spatial-FC layer. In general, MLP models usually set $d = d'$ and $n = n'$ to pursuit efficiency on both feature representation and computation of the entire network.

The spatial-FC layer mentioned above cannot deal with images with diverse shapes, thus is unable to be applied on downstream tasks such as image detection and image segmentation. In order to solve the problem, Cycle-MLP [3] retains the position of each token and introduces Cycle-FC layer to enlarge the receptive field of MLP to cope with downstream tasks while maintaining the computational complexity. Specifically, given the input feature $Z \in \mathbb{R}^{H \times W \times C_{in}}$, the output of Cycle-FC layer is:

$$\text{Cycle-FC}(Z)_{i,j,:} = \sum_{c=0}^{C_i n} Z_{i+\delta_i(c), j+\delta_j(c), c} \cdot W_{c,:}^{cycle}; \tag{6}$$

in which $W^{cycle} \in \mathbb{R}^{C_{in} \times C_{out}}$ is the parameter matrix of the Cycle-FC layer, and $\delta_i(c)$ and $\delta_j(c)$ are defined as:

$$\delta_i(c) = (c \mod S_H) - 1, \quad \delta_j(c) = (\lfloor \frac{c}{S_H} \rfloor \mod S_W) - 1, \tag{7}$$

where $S_H$ and $S_W$ are predefined receptive fields. Besides Cycle-MLP, Wave-MLP [40] represents token as a wave function with amplitude and phase and proposes Wave-FC layer to solve the above problem. AS-MLP [22] avoids the restriction of fixed input size by axially shifting channels of the feature map, and is able to obtain information flow from different axial directions.

Compared to the convolutional operation in CNN and self-attention operation in Transformer, matrix multiplication in MLP is computational friendly to GPUs and has specific advantage on inference time in reality. Benefits from its simple architecture, MLP models can be generalized to various hardware. However, the cumbersome MLP model with a large number of parameters and FLOPs limits its ability to apply to portable devices, which means it is essential to derive a compact MLP model, *e.g.* binary vision MLPs.

# 3 Binary Vision MLPs

In this section, we first analyze the difficulty of binarizing vision MLP through comparing the representation ability of binary features in FC layers and convolutional layers. We then propose a novel binary architecture for vision MLP, which fuses the outputs from the same layer with multi-branch MLP block and those from different layers with a universal shortcut connection to strengthen the representation ability of MLP.

## 3.1 Difficulty of Binarizing Vision MLP

Given the input feature $F \in \mathbb{R}^{C_{in} \times H \times W}$ and the convolutional kernel $K \in \mathbb{R}^{C_{out} \times C_{in} \times K_h \times K_w}$ in which $K_h$ and $K_w$ are the height and width of the kernel, respectively. The **computational complexity** of the convolutional operation is:

$$\mathcal{O}(C_{in} \times C_{out} \times K_h \times K_w \times H \times W). \tag{8}$$

After binarizing the weights and activations, the elements of the output feature map $Y$ is derived from the following equation:

$$Y(o, i, j) = \sum_{c=1}^{C_{in}} \sum_{d_x=-\lfloor \frac{K_w}{2} \rfloor}^{\lfloor \frac{K_w}{2} \rfloor} \sum_{d_y=-\lfloor \frac{K_h}{2} \rfloor}^{\lfloor \frac{K_h}{2} \rfloor} F^b(c, i+d_y, j+d_x) \times K^b(o, c, d_y + \frac{K_h}{2}, d_x + \frac{K_w}{2}), \tag{9}$$

in which $K^b$ and $F^b$ are binarized kernel and input feature using Eq. 1 whose elements are selected from $\{+1, -1\}$. According to Eq. 9, we can easily conclude that each element in $Y$ is chosen from $N+1$ different values from the set $S_b = \{-N, -N+2, ..., N-2, N\}$, in which $N = C_{in} \cdot K_h \cdot K_w$. Specifically, we denote the number $N$ as the **representation ability** of the binary FC layer.

In traditional CNN models such as ResNet [14] and VGGNet [38], convolutional operation with kernel size $3 \times 3$ occupies the network. Efficient-Net [39] uses convolution with kernel size $3 \times 3$ and $5 \times 5$, while the recently proposed RepLKNet [7] expands the kernel size to $31 \times 31$. Different from the CNN models mentioned above, FC layers in MLP model can be treated as convolutional operation with kernel size $1 \times 1$.

Therefore, compared to the convolutional layer with kernel size $k \times k$, FC layer with the same number of input channels and output channels has $1/k^2$ computational complexity and $1/k^2$ representation ability after being binarized. Note that assuming CNN models and MLP models have the same number of input channels and output channels is reasonable. For example, Wave-MLP-S [40] with $30M$ parameters and $4.5G$ FLOPs has four stages with base channel 64-128-320-512, while ResNet-50 with $25.5M$ parameters and $4.1G$ FLOPs has four stages with base channel 64-128-256-512 which are roughly the same.

In order to achieve the same representation ability as convolutional layer, the number of input channels of FC layer must be multiplied by $k^2$. Similarly, the number of output channels should also be scaled up in order to maintain the representation ability of the next FC layer. Hence, the computational complexity is $k^2$ times compared to the convolutional layer with kernel size $k \times k$, which drastically reduce the advantage of binary neural network. To this end, we design a novel binary architecture for vision models (BiMLP) to enhance the representation ability while maintaining compact model complexity. A series of specific design are proposed to achieve this goal, which will be elaborated in the following sections.

## 3.2 BiMLP Architecture

To deal with the above problem, we introduce a mulit-branch binary MLP block (MBB block) with Uni-shortcut operation to enhance the representation capacity of binary MLP models. We also design a novel architecture of downsampling layer for binary MLPs to further reduce the computational complexity while maintaining the accuracy.

**Multi-branch binary MLP block.** To enhance the representation ability of binary vision MLPs with compact complexity, we propose a new binary MLP block containing multiple branches. Note that in MLP-Mixer [43], ResMLP [44] and gMLP [25], an MLP block mainly consists of two parts

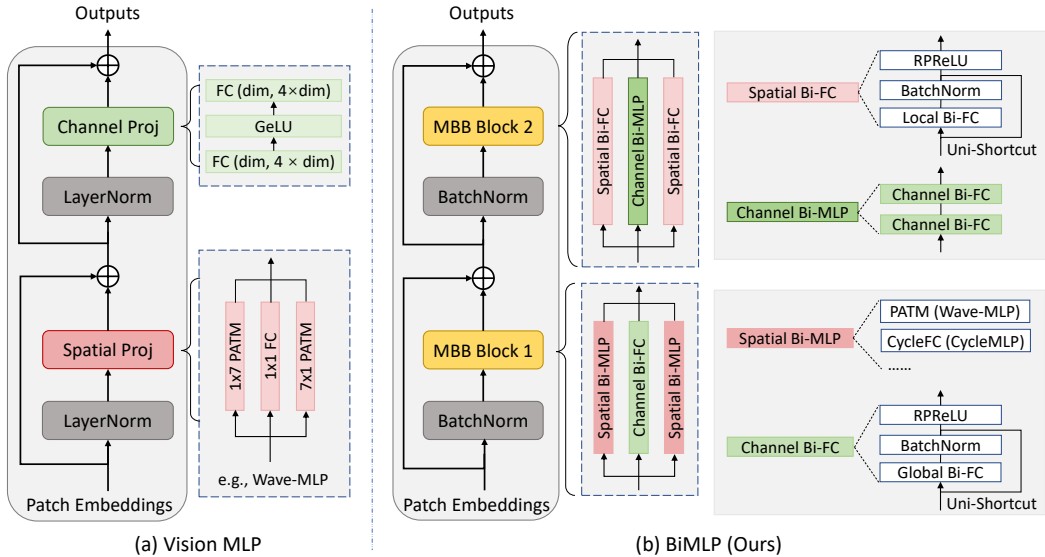

Figure 2: (a) Details of vision MLP blocks, *e.g.*, Wave-MLP [40]. (b) The proposed Multi-branch binary MLP (MBB) blocks for BiMLP.

including the spatial projection part and the channel projection part. For CycleMLP [3] and Wave-MLP [40], the spatial projection is achieved by introducing some **local FC** operations (*e.g.* Cycle-FC, Wave-FC) which are applied on the adjacent tokens to cope with input images with different shapes. Meanwhile, the **global FC** layer (ordinary FC layer) is used for channel mixing. Different from them, we simultaneously introduce spatial projection and channel projection into a single part, and form two different multi-branch binary MLP blocks (MBB blocks) as shown in Fig. 2.

There are four different elements in the MBB blocks, *i.e.*, the Spatial Binary FC, the Spatial Binary MLP, the Channel Binary FC and the Channel Binary MLP. Given the binarized input feature $X^b$, the output of Spatial Binary FC is:

$$Y_{SB\_FC} = \text{RPReLU}(\text{LFC}(\text{BN}(X^b)) + U(X^b)), \quad (10)$$

in which RPReLU($\cdot$) is the activation function introduced in ReActNet [26]. LFC($\cdot$) is the local FC that fuses the spatial information which is realized with different form in previous works. BN($\cdot$) is the ordinary batch normalization, and U($\cdot$) is the proposed Uni-shortcut which will be introduced below in Eq. 12. For Spatial Binary MLP, we utilize the PATM architecture introduced in Wave-MLP [40]. Other forms such as CycleMLP architecture in [3], and axially shifting architecture in AS-MLP [22] can be used as a replacement.

The Channel Binary FC is defined as:

$$Y_{CB\_FC} = \text{RPReLU}(\text{GFC}(\text{BN}(X^b)) + U(X^b)), \quad (11)$$

in which GFC($\cdot$) is the original global FC layer that mixes the information between different channels. Finally, the Channel Binary MLP is the stack of multiple Channel Binary FCs to strengthen the ability of channel mixing.

Given the four different elements introduced above, the first MBB block uses two Spatial Binary MLPs focusing on the height and width of the spatial dimension, and a Channel Binary FC. The second MBB block uses two Spatial Binary FC and a Channel Binary MLP. In this way, the channel projection and spatial projection are used multiple times, while at the same time the first block has a stronger ability for spatial mixing and the second block is good at mixing the channel information. Finally, all the layer normalizations are replaced with batch normalizations, as shown in the ablation study in Tab. 4.

With the above architectures, the representation ability can be recovered with less computational complexity by using multiple branches compared with directly expanding the input and output channels. Generally, $k^2$ branches are needed to get the same representation ability, and the computational complexity is the same compared to the convolutional layer with kernel size $k \times k$. Note that as the number of branches increases, both the representation ability and the computational complexity

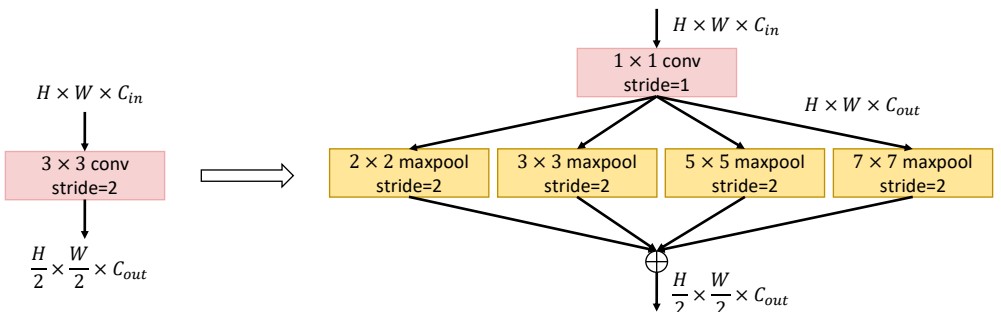

Figure 3: Left: The original downsampling layer used in 32-bit floating point MLP models. Right: The proposed downsampling block for BiMLP.

will increase. Thus, we seek for a balance between the effectiveness and the efficiency. In the following experiments, we use only 3 branches as mentioned above and show that it is enough to obtain competitive results in Tab. 3.

**Uni-shortcut.** Recall that the proposed MBB blocks fuse the information from the same layer. The previous works of binary CNNs show that combining the outputs from different layers is important for the information flow. However, the number of output channels and input channels are usually different in MLP models and the original shortcut connection cannot be directly applied. Since the input channels $c_{in}$ are usually multiple times of the output channels $c_{out}$ (or vise versa), *i.e.* $c_{in} = nc_{out}$ (or $c_{out} = nc_{in}$), $n \in N^+$. Therefore, we propose a universal shortcut (Uni-shortcut) to cope with the following two different situations.

Given the input feature $X^b \in \mathbb{R}^{C_{in} \times H \times W}$ and a FC layer with $c_{in}$ and $c_{out}$ input and output channels, we have:

$$
U(X^b) = \begin{cases} \dfrac{1}{n} \displaystyle\sum_{i=1}^{n} X^b(:, :, \dfrac{i-1}{n}c_{in} : \dfrac{i}{n}c_{in}), & c_{in} = nc_{out}, \\[2mm] concat(\displaystyle\sum_{i=1}^{n} X^b, \ dim = 2), & c_{out} = nc_{in}, \end{cases}
\tag{12}
$$

where the input is averaged based on the channel dimension when $c_{in} = nc_{out}$, and the input is repeated $n$ times and concatenated together to fit the output dimension when $c_{out} = nc_{in}$.

**Downsampling block.** The downsampling layers are not binarized during training and inference for MLP models, otherwise the performance will drastically decrease, as also discussed in prior binary CNNs [27, 26, 33]. This brings the problem that the FLOPs of downsampling layers occupy the whole network since the $3 \times 3$ convolution with stride 2 is a common way of simultaneously reducing the spatial size and changing the number of channels of the feature maps. Thus, similar to [13], we separately handle the spatial and channel dimension by using $1 \times 1$ convolution with stride 1 (FC layer), and maxpooling with diverse kernel size to replace the original convolution, as shown in Fig. 3. By this way, the computational complexity is significantly reduced while the classification performance is maintained.

## 4 Experiments

In this section, we first evaluate the effectiveness of the proposed method on ImageNet dataset. The experimental settings include the dataset statistics, details of the network architectures and the training strategy is introduced in Sec. 4.1. In Sec. 4.2, we compare our method with a series of state-of-the-art binary CNN models in terms of accuracy and FLOPs. The ablation study on each part of the proposed method is conducted in Sec. 4.3.

### 4.1 Experimental Settings

**ImageNet dataset.** The benchmark dataset ImageNet-1k [36] contains over $1.2M$ training images and $50k$ validation images from 1,000 different categories. This is the most commonly used image classification dataset in other binary CNN methods.

Table 1: Details of different architectures of BiMLP. The parameter 'dim' indicates the dimension of features, and 'ratio' is the expand ratio of the mid layer of MLP. FLOPs is calculated on the layers that are not binarized, BOPs is calculated on binarized layer and OPs = BOPs / 64 + FLOPs. The computational complexity is calculated based on the input shape $224 \times 224$.

| | Output size | BiMLP-S | BiMLP-M |
|---|---|---|---|
| Stage 1 | $\frac{H}{4} \times \frac{W}{4}$ | $\begin{bmatrix} \text{dim} = 64 \\ \text{ratio} = 4 \end{bmatrix} \times 2$ | $\begin{bmatrix} \text{dim} = 64 \\ \text{ratio} = 4 \end{bmatrix} \times 2$ |
| Stage 2 | $\frac{H}{8} \times \frac{W}{8}$ | $\begin{bmatrix} \text{dim} = 128 \\ \text{ratio} = 4 \end{bmatrix} \times 2$ | $\begin{bmatrix} \text{dim} = 128 \\ \text{ratio} = 4 \end{bmatrix} \times 3$ |
| Stage 3 | $\frac{H}{16} \times \frac{W}{16}$ | $\begin{bmatrix} \text{dim} = 320 \\ \text{ratio} = 4 \end{bmatrix} \times 4$ | $\begin{bmatrix} \text{dim} = 320 \\ \text{ratio} = 4 \end{bmatrix} \times 10$ |
| Stage 4 | $\frac{H}{32} \times \frac{W}{32}$ | $\begin{bmatrix} \text{dim} = 512 \\ \text{ratio} = 4 \end{bmatrix} \times 2$ | $\begin{bmatrix} \text{dim} = 512 \\ \text{ratio} = 4 \end{bmatrix} \times 3$ |
| FLOPs ($\times 10^8$) | | 1.21 | 1.21 |
| BOPs ($\times 10^9$) | | 2.25 | 4.32 |
| OPs ($\times 10^8$) | | 1.56 | 1.88 |

**Network architecture.** We utilize the architecture proposed in [40] and replace the MLP block with the proposed MBB block with Uni-shortcut (Fig. 2(b)) as well as the replacement of downsampling blocks (Fig. 3). Details of the architectures of BiMLP models are specified in Tab. 1.

**Training details.** Following prior methods [27, 26, 30], we use a two step training strategy. In the first training step, we use the full precision MLP model as the teacher model and the network with the same architecture but binary activations as the student model. A knowledge distillation loss is used to facilitate the training of student network:

$$L = \alpha \sum_{i=1}^{n} L_{kl}(\mathbf{y}_s, \mathbf{y}_t) + (1 - \alpha) \sum_{i=1}^{n} L_{ce}(\mathbf{y}_s, \mathbf{y}_{gt}), \tag{13}$$

in which $L_{kl}(\cdot)$ and $L_{ce}(\cdot)$ are the KL divergence and cross-entropy loss, $\mathbf{y}_s$ and $\mathbf{y}_t$ are the output probability of the student and teacher model, $\mathbf{y}_{gt}$ is the one-hot ground-truth probability and $\alpha$ is the trade-off hyper-parameter ($\alpha = 0.9$ in the following experiments). For the second training step, the full precision ResNet-34 is used as the teacher network. The binarized MLP model (both weights and activations) is used as the student model and the corresponding weights are initialized by the training results from the first step.

In both steps, the student models are trained for 300 epochs using AdamW [28] optimizer with momentum of 0.9 and weight decay of 0.05. We start with the learning rate of $1 \times 10^{-3}$ and a cosine learning rate decay scheduler is used during training. We use NVIDIA V100 GPUs with a total batchsize of 1024 to train the model with Mindspore [17]. The commonly used data-augmentation strategies such as Cut-Mix [51], Mixup [52] and Rand-Augment [5] are used. The first layer, the last layer and the downsampling layers are not binarized during training and inference.

## 4.2 Experimental Results on ImageNet

In this section, we compare the proposed model with the binary CNN models derived from other state-of-the-art methods on the ImageNet-1k dataset, including Dorefa-Net [53], XNOR-Net [33], ABCNet [24], Bireal-Net [27], Real2bin [30], MeliusNet [2], ReActNet [26], *etc.* As shown in Tab. 2, the proposed BiMLP models achieve competitive performance compared to the state-of-the-art binary CNN models. We can see that BiMLP-S achieves 70.0% top-1 accuracy and 89.6% top-5 accuracy with 0.156G OPs, which surpasses the most recent methods such as MeliusNet-42 by 0.8% and 1.3% with less than a half OPs, and is comparable to ReActNet-B. Meanwhile, BiMLP-M achieves 72.7% top-1 accuracy and 91.1% top-5 accuracy with only 0.188G OPs, which has 12.1% less OPs than ReActNet-C and is 1.3% better on top-1 accuracy. The above results show the superiority of the proposed BiMLP.

Table 2: Experimental results on ImageNet-1k using different binary CNN models and the proposed BiMLP. Bit-width (W/A) indicates the bit length of weights and activations. FLOPs is calculated on the full-precision layers, BOPs is calculated on binarized layers and OPs = BOPs / 64 + FLOPs.

| Methods | Bit-width (W/A) | FLOPs ($\times 10^8$) | BOPs ($\times 10^9$) | OPs ($\times 10^8$) | Top-1 Acc (%) | Top-5 Acc (%) |
|---|---|---|---|---|---|---|
| XNOR-Net [33] | 1/1 | 1.41 | 1.70 | 1.67 | 51.2 | 73.2 |
| Dorefa-Net [53] | 1/1 | - | - | - | 52.5 | 67.7 |
| ABCNet [24] | 1/1 | - | - | - | 42.7 | 67.6 |
| Bireal-Net-18 [27] | 1/1 | 1.39 | 1.68 | 1.63 | 56.4 | 79.5 |
| Bireal-Net-34 [27] | 1/1 | 1.39 | 3.53 | 1.93 | 62.2 | 83.9 |
| UAD-BNN-18 [18] | 1/1 | - | - | - | 57.2 | 80.2 |
| UAD-BNN-34 [18] | 1/1 | - | - | - | 62.8 | 84.5 |
| RBNN-18 [23] | 1/1 | - | - | - | 59.9 | 81.9 |
| RBNN-34 [23] | 1/1 | - | - | - | 63.1 | 84.4 |
| Real2bin [30] | 1/1 | 1.56 | 1.68 | 1.83 | 65.4 | 86.2 |
| ReCU [50] | 1/1 | - | - | - | 66.4 | 86.5 |
| FDA-BNN [47] | 1/1 | - | - | - | 66.0 | 86.4 |
| LCR-BNN-18 [37] | 1/1 | - | - | - | 59.6 | 81.6 |
| LCR-BNN-34 [37] | 1/1 | - | - | - | 63.5 | 84.6 |
| LCR-BNN-ReAct [37] | 1/1 | - | - | - | 69.8 | 85.7 |
| Bi-half-18 [20] | 1/1 | - | - | - | 60.4 | 82.9 |
| Bi-half-34 [20] | 1/1 | - | - | - | 64.2 | 85.4 |
| AdaBin-18 [45] | 1/1 | 1.41 | 1.69 | 1.67 | 66.4 | 86.5 |
| AdaBin-ReAct [45] | 1/1 | - | - | 0.88 | 70.4 | - |
| AdaBin-Melius59 [45] | 1/1 | - | - | 5.27 | 71.6 | - |
| MeliusNet-22 [2] | 1/1 | 1.35 | 4.62 | 2.08 | 63.6 | 84.7 |
| MeliusNet-29 [2] | 1/1 | 1.29 | 5.47 | 2.14 | 65.8 | 86.2 |
| MeliusNet-42 [2] | 1/1 | 1.74 | 9.69 | 3.25 | 69.2 | 88.3 |
| MeliusNet-59 [2] | 1/1 | 2.45 | 18.3 | 5.25 | 71.0 | 89.7 |
| ReActNet-B [26] | 1/1 | 0.44 | 4.69 | 1.63 | 70.1 | - |
| ReActNet-C [26] | 1/1 | 1.40 | 4.69 | 2.14 | 71.4 | - |
| **BiMLP-S (Ours)** | 1/1 | 1.21 | 2.25 | **1.56** | **70.0** | **89.6** |
| **BiMLP-M (Ours)** | 1/1 | 1.21 | 4.32 | **1.88** | **72.7** | **91.1** |

Table 3: Experimental results of using different branches in MBB blocks. The experiments are conducted using BiMLP-S model on ImageNet-1k dataset, and setting 3 is used as the baseline architecture which is the same as Fig. 2(b).

| Setting | MBB Block 1 | | MBB Block 2 | | OPs ($\times 10^8$) | Top-1 Acc (%) |
|---|---|---|---|---|---|---|
| | #$S_1$ | #$C_1$ | #$S_2$ | #$C_2$ | | |
| 1 | 4 | 0 | 0 | 2 | 1.60 | 69.2 |
| 2 | 0 | 2 | 4 | 0 | 1.53 | 68.9 |
| 3 | 2 | 1 | 2 | 1 | 1.56 | 70.0 |
| 4 | 4 | 1 | 2 | 2 | 1.86 | 70.8 |
| 5 | 2 | 2 | 4 | 1 | 1.78 | 70.5 |

## 4.3 Ablation Studies

In this section, we demonstrate the effectiveness of each part of the proposed method by conducting a series of ablation studies.

**Different branches in MBB blocks.** Firstly, we show the experimental results of using different branches in Tab. 3. The #$S_i$ indicates the number of Spatial Binary FC (MLP) in the MBB block $i$ and #$C_i$ is the number of Channel Binary FC (MLP) in the corresponding block. Note that we fuse the information along height and width of the feature map with different Spatial Binary FC (MLP), thus the #$S_i$ should be even numbers. During the ablation study we keep $(\#S_1 + \#S_2)/(\#C_1 + \#C_2) = 2$ so that the ability of fusing information of different dimensions is roughly the same.

The experiments are conducted on BiMLP-S model. As shown in Tab. 3, when the MBB blocks focus on only one dimension in setting 1 and 2, the top-1 accuracy decreases from 70.0% to 69.2%

Table 4: Different forms of normalization layer and activation function. The experiments are conducted using BiMLP-S model on ImageNet-1k dataset.

| Normalization | LN | LN | LN | LN | BN | BN | BN | BN |
|---|---|---|---|---|---|---|---|---|
| Activation | GeLU | ReLU | PReLU | RPReLU | GeLU | ReLU | PReLU | RPReLU |
| Top-1 Acc (%) | 68.2 | 67.8 | 68.4 | 69.2 | 68.5 | 68.1 | 69.4 | **70.0** |

Table 5: Experimental results of using different form of residual connections. The experiments are conducted on ImageNet-1k dataset with BiMLP-S model.

| | Top-1 Acc (%) |
|---|---|
| w/o shortcut | 68.3 |
| w/ Uni-shortcut | **70.0** |
| w/ shortcut | 69.1 |

Table 6: Results of using different downsampling layers. 'Original' indicates the $3 \times 3$ convolution with stride 2, and 'proposed' means the proposed downsampling block as shown in Fig. 3 right.

| Setting | OPs ($\times 10^8$) | Top-1 Acc (%) |
|---|---|---|
| Original | 2.65 | 70.3 |
| Proposed | 1.56 | 70.0 |

and 68.9%, which shows that mixing the information from different dimensions in a single block is important. In setting 4 and 5, more branches are used in each block compared to setting 3 (baseline), and only improves the accuracy by 0.8% and 0.5%, while increases 19.2% and 14.1% OPs. Thus, we use setting 3 for all the experiments considering the balance between the effectiveness and efficiency.

**Normalization and activation.** Different forms of normalization layer and activation function are compared. The candidates of normalization layer include Layer normalization (LN) and batch normalization (BN), and the candidates of activation functions are GeLU, ReLU, PReLU and RPReLU. As the results shown in Tab. 4, the combination of BN and RPReLU achieves the best accuracy, and is applied to all other experiments.

**Shortcut connection.** We also study the effectiveness of the proposed Uni-shortcut. Three different setting are used in this ablation study. The first is using the MBB block without any residual connection. The second is using the proposed Uni-shortcut in MBB blocks. The third is using the traditional shortcut in MBB blocks. In this circumstances, only those input-output pairs who have the same shape will be connected.

As shown in Tab. 5, compared to the MBB block without using any residual connection and using the traditional shortcut, Uni-shortcut improves the top-1 accuracy by 1.7% and 0.9% which shows the priority of the proposed method.

**Downsampling layer.** We compare the proposed downsampling block using the FC layer together with multi-branch maxpooling to the original downsampling layer using $3 \times 3$ convolution with stride 2 in Tab. 6. The result shows that we can significantly reduce the computational complexity from 0.265G to 0.156G (41.1% decrease) with only 0.3% drop of top-1 classification accuracy.

**Hyper-parameters.** As shown in Eq. 13, $\alpha$ is used as a hyper-parameter to balance the loss function between learning from the output probability of teacher model and learning from the ground-truth

Table 7: Results of using different binarization methods on ImageNet-1k dataset. The 'FP32' model indicates the Wave-MLP model, and '1-bit' model uses sign function and STE rule to binarize the FP32 model.

| Model | Top-1 Acc(%) |
|---|---|
| FP32 | 80.1 |
| 1-bit | 55.3 |
| + Dorefa-Net [53] | 58.4 |
| + XNOR-Net [33] | 59.1 |
| + ReActNet [26] | 63.2 |
| + Ours | **70.0** |

Figure 4: Top-1 acc vs. $\alpha$ on ImageNet val set.

one-hot label. We report the accuracy on ImageNet-1k validation set in Fig. 4 and show that $\alpha = 0.9$ yields the best result.

**Comparisons of different binarization methods.** We compare the proposed method to other binarization methods proposed for CNN models in Tab. 7. The 1-bit model is to directly binarize the FP32 Wave-MLP model using the sign function and the STE rule. The experimental results show that our method can significantly outperform other binarization methods that are directly applied to MLP model.

**Overall benefit of modifying the architecture.** We conduct an ablation study to show the overall benefit of modifying the architecture. The accuracy gaps between FP32 and the binary version of the original MLP and the modified MLP are reported in Tab. 8. We can see that the FP32 variant of the modified model achieves roughly the same performance with original FP32 model. Meanwhile, our proposed BiMLP architecture outperforms the 1bit Wave-MLP with a large margin, which justifies the effectiveness of the proposed binary vision MLP architecture.

Table 8: Comparisons of FP32 and binary variants for original and our proposed architectures on ImageNet-1k.

| Model | Wave-MLP | BiMLP |
|-------|----------|-------|
| FP32  | 80.1     | 79.9  |
| 1-bit | 63.2     | 70.0  |

## 5   Conclusion

In this paper, we propose a new paradigm with specific architecture design for binarizing the vision MLP models. We point out that compared to binarizing the convolutional operation with large kernel size in CNN models, binarizing the FC layers yields a MLP model with poor representation ability. Simply increase the number of channels will fix this problem, but the computational complexity is quadratically increased. Thus, we propose a novel multi-branch binary MLP block (MBB block) with universal shortcut (Uni-shortcut) that can recover the representation ability while maintaining the computational complexity. We also redesign the downsampling layer that can significantly reduce the OPs of the binary MLP model while at the same time keep the performance roughly unchanged. The experimental results on ImageNet-1k dataset demonstrate the effectiveness of the proposed method, and we achieve a comparable performance with the state-of-the-art binary CNN models.

## Acknowledgments

We gratefully acknowledge the support of MindSpore, CANN (Compute Architecture for Neural Networks) and Ascend AI Processor used for this research.

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
