# OpenReview forum: "BiMLP: Compact Binary Architectures for Vision Multi-Layer Perceptrons"
_NeurIPS.cc/2022/Conference — NeurIPS 2022 Accept_

### Official Review · Reviewer_NgiC · 2022-07-04

**Rating:** 7
**Confidence:** 4
**Soundness:** 3 good
**Presentation:** 4 excellent
**Contribution:** 3 good

**Summary:**

This paper proposes a new architecture for binarizing the vision MLP model and mitigates the gap between the FP32 model and the 1bit version. It designs a novel MBB block with a universal shortcut to strengthen the representation ability of the binary FC layer, and redesigns the downsampling layers to reduce the computational complexity while maintaining the classification performance. The experimental results on ImageNet dataset show that the performance of the proposed binary MLP model can exceed the state-of-the-art binary CNN model, and the ablation study demonstrates the effectiveness of each proposed module.

**Questions:**

See weaknesses above. I suggest the author gives an example to prove that the assumption mentioned above is true in reality. Otherwise, it may influence the conclusion of the analysis.

Is 1x1 conv in the proposed downsampling block in Figure 3 binarized or remain FP32?

The font in Figure 2 should be unified.


**Limitations:**

Yes.

**Strengths And Weaknesses:**

Strengths:
- The analysis of computational complexity and representation ability of binary conv layer and binary FC layer is inspiring. The author points out that the representation ability of a binary conv/FC layer is related to the kernel size and the input channel, which helps the design of the following MBB block. I think this analysis is also helpful for designing an effective binary CNN module.
- The use of multi-branch block is simple yet reasonable and effective. The authors uniformly distribute computation power on different dimensions, and substantially increase the representation ability while maintaining the computational complexity of binary FC layers, compared to binary conv layer with large kernel size.
- The experimental results show the effectiveness of the proposed architecture, and it outperforms the SOTA binary CNN model ReActNet.

Weaknesses
- This paper should conduct an ablation study to show the overall benefit of modifying the architecture. For example, the accuracy gap between FP32 and the binary version of the original MLP and the modified MLP should be reported.
- When analyzing the representation ability and computational complexity, this paper assumes that the number of input channel and output channel of binary conv in CNN and binary FC in MLP are the same, is it true in reality?

---

> ### Author Response · Authors · 2022-08-01
> **Rebuttal to Reviewer NgiC**
>
> Thanks for your constructive comments and support.
>
> ***Q1: This paper should conduct an ablation study to show the overall benefit of modifying the architecture. For example, the accuracy gap between FP32 and the binary version of the original MLP and the modified MLP should be reported.***
>
> A1: We conduct experiments with Bi-MLP-S on ImageNet dataset, and the experimental results are shown below:
>
> |Model|Top-1 Acc|
> |-|-|
> |original model FP32| 80.1|
> |original model 1bit|63.2|
> |modified model FP32|79.9|
> |modified model 1bit|70.0|
>
> Note that the results of original model FP32, original model 1bit and modified model 1bit are already given in Table 7 line 1, 5 and 6 in the original paper. We further conduct experiments on modified model FP32, and show that the result is roughly the same with original model FP32.
>
> This is reasonable, since the FP32 model does not have the trouble of poor representation ability as 1bit model has, and thus the proposed modules will have less effect on the final performance of the full precision model.
>
> ***Q2: When analyzing the representation ability and computational complexity, this paper assumes that the number of input channel and output channel of binary conv in CNN and binary FC in MLP are the same, is it true in reality?***
>
> A2: This is a really good and interesting question that two reviewers care about. We compare WaveMLP-S (the full-precision version of Bi-MLP-M) with 30M parameters and 4.5G FLOPs and the traditional ResNet-50 with 25.5M parameters and 4.1G FLOPs, and found that the number of channels between the two models are roughly the same, as shown below:
>
> ResNet-50
>
> |stage1 ($\times3$)|stage2 ($\times4$)|stage3 ($\times6$)|stage4 ($\times3$)|
> |-|-|-|-|
> |1x1, 64|1x1, 128|1x1, 256|1x1, 512|
> |3x3, 64|3x3, 128|3x3, 256|3x3, 512|
> |1x1, 256|1x1, 512|1x1, 1024|1x1, 2048|
>
> WaveMLP-S
>
> |stage1 ($\times2$)|stage2 ($\times3$)|stage3 ($\times10$)|stage4 ($\times3$)|
> |-|-|-|-|
> |dim=64|dim=128|dim=320|dim=512|
> |ratio=4|ratio=4|ratio=4|ratio=4|
>
> We can see that both models have 4 stages. ResNet-50 with base channel $64-128-256-512$, and WaveMLP-S with base channel $64-128-320-512$. ResNet-50 expand the channel by 4x at the end of each stage, while WaveMLP-S expand the channel by 4x in the middle of the stage and then shrink back to the base channel. Thus, we can say that the number of channels in MLP and CNN models are roughly the same, which means that the claim of representation ability can roughly hold. Details of analysis will be added in the final version of the paper.
>
> ***Q3: Is 1x1 conv in the proposed downsampling block in Figure 3 binarized or remain FP32?***
>
> A3: It remains FP32 during the experiments.
>
> ***Q4: The font in Figure 2 should be unified.***
>
> A4: Thanks for your suggestion, we will unify the font in Figure 2 in the final version.

---

> > ### Comment · Reviewer_NgiC · 2022-08-09
> > **After response**
> >
> > Thanks for your response. My concerns are well addressed in this response. Thus, I keep my original rate to accept this paper.

---

### Official Review · Reviewer_qm2m · 2022-07-09

**Rating:** 6
**Confidence:** 4
**Soundness:** 3 good
**Presentation:** 3 good
**Contribution:** 3 good

**Summary:**

This paper presents a binary MLP model. Specifically, it designs the binary block with multiple branches (MBB blocks) and uses the shortcut to encourage the information flow. For MBB blocks, the authors proposed to  form two different multi-branch blocks by using spatial projection and channel projection simultaneously.

**Questions:**

1. Any intuition about using the multi-branch blocks and the down-sampling block with multiple max-pooling branches?
2. What is the original size of the MLP network
3. Can the authors compare the FLOPs as well as the model size? Because MLP networks are computational efficient, but the #parameters might be much greater than CNNs.


**Limitations:**

The authors claimed the FC layers can be viewed as 1x1 Conv and thus is limited in representation ability. However, the number of channels in FC layers are usually much larger than 1x1 Convolutions used in the CNNs. In that case this claim doesn't quite hold.

**Strengths And Weaknesses:**

(+) The paper is well-written. The result shows it achieves even better accuracy than the state-of-the-art binary CNNs, which is impressive.

(-) I didn't see a strong motivation for using multi-branch blocks.

---

> ### Author Response · Authors · 2022-08-01
> **Rebuttal to Reviewer qm2m**
>
> Thanks for your constructive comments and support.
>
> ***Q1: Any intuition about using multi-branch blocks and down-sampling block with multiple max-pooling branches?***
>
> A1: The motivation comes from the architecture difference between MLP and CNN. The only difference between FC layer (basic element in MLP) and conv layer (basic element in CNN) is that FC layer can be treated as convolution with kernel size 1, while conv layer in CNN always have larger kernel size. As shown in Line 144 in original paper, the representation ability of binary FC and conv layer is related to the kernel size (N=C_in * K_h * K_w), and binary FC layer tend to have less representation ability due to the small kernel size and yields poor performance, as shown in the table below.
>
> |Network|Kernel Size|Performance drop|
> |-|-|-|
> |WaveMLP|1|22%|
> |ResNet-18|3|17%|
> |AlexNet|11 & 5|13%|
>
> Note that the larger the kernel size, the less performance drop between 1bit network and full-precision network. Thus, we need to increase the representation ability of MLP.
>
> In order to make the representation ability of FC layer (1x1 conv) to be the same as conv layer (kxk conv), there are two different ways. The first is to increase the input channel. Note that output channels should also be scaled up in order to maintain the representation ability (RA) of the next FC layer (the number of output channel of current layer is the number of the input channel of next layer). Thus, the computational complexity (CC) will be drastically increased, as shown in the table below.
>
> ||in_channel|out_channel|kernel_size|CC|RA|
> |-|-|-|-|-|-|
> |bi-FC layer|$C_{in}$|$C_{out}$|$1\times1$|$1$|$1$|
> |bi-conv layer|$C_{in}$|$C_{out}$|$k\times k$|$k^2$|$k^2$|
> |bi-FC layer with more channel|$k^2C_{in}$|$k^2C_{out}$|$1\times1$|$k^4$|$k^2$|
>
> Thus, we use multi-branch blocks to increase the representation ability while maintain the computational complexity, as shown below.
>
> ||branch_num|in_channel|out_channel|kernel_size|CC|RA|
> |-|-|-|-|-|-|-|
> |bi-FC layer|$1$|$C_{in}$|$C_{out}$|$1\times 1$|$1$|$1$|
> |bi-conv layer|$1$|$C_{in}$|$C_{out}$|$k\times k$|$k^2$|$k^2$|
> |bi-FC layer with more channel|$1$|$k^2C_{in}$|$k^2C_{out}$|$1\times1$|$k^4$|$k^2$|
> |bi-FC layer with more branches|$k^2$|$C_{in}$|$C_{out}$|$1\times1$|$k^2$|$k^2$|
>
> The intuition of downsampling block is simple. Original downsampling layers (Figure 3 left) occupy the OPs of the whole binary network, and directly binarize them yield severe performance drop. Thus, we separate the changing of spatial size and channel number with maxpooling and 1x1 conv, and reduce the OPs while keeping the performance.
>
> ***Q2: What is the original size of the MLP network.***
>
> A2: We use WaveMLP [r1] as baseline architecture, and original size of WaveMLP-T and WaveMLP-S (correspond to Bi-MLP-S and Bi-MLP-M) are 17M and 30M.
>
> ***Q3: Can authors compare FLOPs as well as model size? Because MLP networks are computational efficient, but the parameters might be greater than CNNs.***
>
> A3: The FLOPs are shown in Table 2 in original paper. Model size are shown below. Since only the code of ReActNet-A is released, we can only compute the parameters of ReActNet-A.
>
> |Binary Model|#params|Full precision model|model_size|
> |-|-|-|-|
> |ReActNet-A|1.95M|ReActNet-A (FP)|29M|
> |Bi-MLP-S|1.45M|WaveMLP-T|17M|
> |Bi-MLP-M|2.05M|WaveMLP-S|30M|
>
> Note we have less parameters since ReActNet-A is deeper (14 blocks) while Bi-MLP is shallower (10 blocks for Bi-MLP-S).
>
> ***Q4: The authors claimed the FC layers can be viewed as 1x1 Conv and thus is limited in representation ability. However, the number of channels in FC layers are usually much larger than 1x1 Convolutions used in the CNNs. In that case this claim doesn't quite hold.***
>
> A4: This is a really good and interesting question. We compare WaveMLP-S (full-precision version of Bi-MLP-M) with 30M parameters and 4.5G FLOPs and traditional ResNet-50 with 25.5M parameters and 4.1G FLOPs, and found that the number of channels between two models are roughly the same:
>
> ResNet-50
>
> |stage1 ($\times3$)|stage2 ($\times4$)|stage3 ($\times6$)|stage4 ($\times3$)|
> |-|-|-|-|
> |1x1, 64|1x1, 128|1x1, 256|1x1, 512|
> |3x3, 64|3x3, 128|3x3, 256|3x3, 512|
> |1x1, 256|1x1, 512|1x1, 1024|1x1, 2048|
>
> WaveMLP-S
>
> |stage1 ($\times2$)|stage2 ($\times3$)|stage3 ($\times10$)|stage4 ($\times3$)|
> |-|-|-|-|
> |dim=64|dim=128|dim=320|dim=512|
> |ratio=4|ratio=4|ratio=4|ratio=4|
>
> Both models have 4 stages. ResNet-50 with base channel $64-128-256-512$, and WaveMLP-S with base channel $64-128-320-512$. ResNet-50 expand the channel by 4x at the end of each stage, while WaveMLP-S expand the channel by 4x in the middle of the stage and then shrink back to the base channel. Thus, we can say that the number of channels in MLP and CNN models are roughly the same, which means that the claim of representation ability can roughly hold. Details of analysis will be added in the final version of the paper.
>
> [r1] An Image Patch is a Wave: Phase-Aware Vision MLP. CVPR 2022.

---

> > ### Comment · Reviewer_qm2m · 2022-08-09
> > **Post-rebuttal**
> >
> > Thanks authors for the detailed response. It addressed all my concerns. I will maintain my original score.

---

### Official Review · Reviewer_BkjE · 2022-07-10

**Rating:** 5
**Confidence:** 3
**Soundness:** 3 good
**Presentation:** 3 good
**Contribution:** 3 good

**Summary:**

In this paper, the authors first analyze the difficulty of binarizing MLP models due to the amount of FC layers (1x1 conv) that limit the ability for feature representation. Then, the authors propose mulit-branch binary MLP block to improve the accuracy and reduce the OPs compared to the stoa methods.

**Questions:**

I don't have specific questions and suggestions for the authors. I tend to accept this paper but I don't have much experience on model quantization and binary networks. I probably change my rating after the rebuttal and discussion phase.

**Limitations:**

The authors didn't address their limitations and potential negative social impact.

**Strengths And Weaknesses:**

Strengths:

+Clear motivation and writing

The authors analyze the difficulty of binarizing MLP models and propose their solutions with MBB blocks. The writing is overall clear and straightforward.

+Strong experiment results

The authors conduct experiments based on MLP models and achieve better performance with less OPs.

Weaknesses:

-Missing related works

For the down-sampling layers proposed in this paper, some related works have been using such architecture like [1] but missing in the related work part.

[1] Spatial Pyramid Pooling in Deep Convolutional Networks for Visual Recognition, TPAMI 2015

---

> ### Author Response · Authors · 2022-08-01
> **Rebuttal to Reviewer BkjE**
>
> Thanks for your support.
>
> ***Q1: For the down-sampling layers proposed in this paper, some related works have been using such architecture like [1] but missing in the related work.***
>
> A1: Thanks for your suggestion, [1] will be cited and discussed in the final version of the paper.
>
> ***Q2: The authors didn't address their limitations and potential negative social impact.***
>
> A2:  BNN models are usually used on mobile devices with CPU, NPU or FPGA, and we do not compare the actual inference time of different BNN models on these mobile devices.

---

> > ### Comment · Reviewer_BkjE · 2022-08-10
> > **Post-rebuttal**
> >
> > Thanks for your response. I don't have more questions and will maintain my rating.

---

### Official Review · Reviewer_orUW · 2022-07-11

**Rating:** 4
**Confidence:** 4
**Soundness:** 2 fair
**Presentation:** 2 fair
**Contribution:** 2 fair

**Summary:**

In this paper, the author combines channel projection, spatial projection, and the shortcut into a new architecture. The proposed is only validated on one dataset.
Overall, the novelty of the proposed method is weak. The insufficient experiments can support the effectiveness of the proposed method. The paper is not well organized. Two many preliminaries are given, and some critical definitions are missing.


**Questions:**

1. The advantage of ‘Ops’ is benefit from the downsampling layers. Are downsampling layers also added to the SOTA methods [17][2]?
2. In the ablation study, the short-cut, BN+RPReLU, different MBB block settings achieve about 0.9%, 0.6%, and 0.8% increase, respectively. The overall framework only achieves 70% (Bi-MLP-S) and 72.2% (Bi-MLP-M) accuracy. The SOTA method (ReActNet) achieve 71.4% accuracy. So, which part brings the fundamental increase compared with SOTA methods?
3. What’s the meaning of Bi-MLP-S and Bi-MLP-M?
4. In page 7, the author mentioned that “The commonly used data-augmentation strategies such
 as Cut-Mix [33], Mixup [34] and Rand-Augment [5] are used. The first layer, the last layer and the downsampling layers are not binarized during training and inference.” Does the SOTA method also adopt the same data-augmentation strategy? The 1.3% increase is benefit from the data-augmentation strategy. The author is suggested to add the ablation study on the setting with and without data-augmentation strategy.


**Ethics Review Area:**

["I don’t know"]

**Limitations:**

1. The improved performance is limited. The proposed method only achieves 1.3% increase.
2. From Table3, we can see that setting 3 only achieves 0.8% increase than setting 1, which is a simple combination of special projection and channel projection. So I suspect that the main contribution of the combination of special projection and channel projection is not really useful.
3. The proposed method is only validated on one dataset.
4. The proposed method is the integration of existing techniques.
5. The author only gives the Ops performance with block setting from an experimental viewpoint. The author is suggested to add the theoretical analysis of inference performance with different block settings.


**Strengths And Weaknesses:**

Advantages:
1: A novel binary architecture is proposed and achieves SOTA performance.

Disadvantages:
1. The novelty is weak. The author combines the channel projection, spatial projection, and the shortcut into a new module. The core techniques channel projection, spatial projection, and the shortcut are existing techniques. The author modified them to be suitable for the binary vision MLP.
2. The improved performance is limited. Too many techniques (short-cut, BN+RPReLU, different MBB block settings a) are combined into one model, which only achieves 1.3% increase (71.4%->72.7%).
3. The comparison experiment is not sufficient. Most of the compared methods are before 2020. The author should compare the proposed methods with some SOTA methods published in 2021 and 2022.
4. The proposed method is only validated on one dataset.
5. The paper is not well organized, and some important definitions are not given. For example, there are two pages for the preliminaries and Vision MLP Models before the proposed method. However, there are only two pages for the proposed method. The definition of two important methods Bi-MLP-S and Bi-MLP-M are missing.
6. The devised new framework is not useful. The core contribution combining special projection and channel projection doesn’t achieve a large increase (less than 0.8%) than the simple combination of special projection and channel projection (setting 1 in Table 3).

---

> ### Author Response · Authors · 2022-08-01
> **Rebuttal to Reviewer orUW (Weaknesses part)**
>
> Thanks for your constructive comments.
>
> ***Q1.1: The novelty is weak. The author combines channel projection, spatial projection, and shortcut into a new module. The core techniques channel projection, spatial projection, and shortcut are existing techniques. The author modified them to be suitable for binary vision MLP.***
>
> A1.1: The channel projection, spatial projection and shortcut are not our core contributions in this paper. As shown in Table 7, directly binarizing MLP with channel projection and spatial projection yields quite bad performance (55.3% top-1 accuracy). We thus provide thorough analysis for the main difficulty of binarizing MLP models, and propose a series of modules for better binary MLP architecture. Besides, directly applying shortcut used in ReActNet to binarize MLP only yields an accuracy of 63.2%, which is about 7% worse than the proposed method.
>
> To be more clearly, our core motivation is to enhance the representation ability (Line 145) in binary MLP without much increase of the computational complexity (Line 139), which is the main problem when binarizing MLP. Based on this motivation, we propose the Multi-branch binary MLP block, Uniform shortcut and downsampling block which are all used to enhance the representation ability while keeping the computational complexity.
>
> ***Q1.2: The improved performance is limited. Too many techniques (short-cut, BN+RPReLU, different MBB block settings) are combined into one model, which only achieves 1.3% increase  (71.4%->72.7%).***
>
> A1.2: First of all, the ordinary shortcut and BN+RPReLU are also used in ReActNet. In order to achieve a fair comparison to the SOTA method, we use these techniques in our experiments. In fact, 1.3% top-1 accuracy increase on ImageNet dataset is non-negligible, and we have 12.1% less OPs than the competitor. We list the improved performance to their own competitors stated in several recently published papers.
>
> |paper|conference|improved Top-1 Acc|
> |-|-|-|
> |ReCU [r1]|ICCV 21|+0.5|
> |FDA-BNN [r2]|NeurIPS 21|+0.5|
> |AdaSTE [r3]|CVPR 22|+0.3|
> |UAD-BNN [r4]|CVPR 21|+0.9|
> |LCR-BNN [r5]|ECCV 22|+0.4|
> |Equal-Bits-BNN [r6]|AAAI 22|+1.0|
>
>
> ***Q1.3: The comparison experiment is not sufficient. Most of compared methods are before 2020. The author should compare BiMLP with some SOTA methods published in 2021 and 2022.***
>
> A1.3: ReActNet is a strong baseline on ImageNet dataset. Thanks for your suggestion and in below we include more recently published BNN papers and show that we still achieve SOTA result.
>
> |paper|conference|top-1 acc on ImageNet| top-1 acc on tiny ImageNet|
> |-|-|-|-|
> |ReCU [r1]|ICCV 21|66.4|-|
> |FDA-BNN [r2]|NeurIPS 21|66.0|-|
> |AdaSTE [r3]|CVPR 22|-|54.9|
> |UAD-BNN [r4]|CVPR 21|62.8|-|
> |LCR-BNN [r5]|ECCV 22|69.8|-|
> | Equal-Bits-BNN [r6]|AAAI 22|60.4|-|
> |Bi-MLP (ours)|-|72.7|-|
>
> Methods [r1]-[r6] will be added in final version of the paper for comparison.
>
> ***Q1.4: The proposed method is only validated on one dataset.***
>
> A1.4: The most commonly used datasets in the area of BNN are CIFAR-10, CIFAR-100 and ImageNet, in which ImageNet is the most convincing and challenging dataset to show the effectiveness of a method. We conduct extensive experiments and ablation studies on ImageNet, which we believe is sufficient enough to demonstrate the performance of our method.
>
> ***Q1.5: The paper is not well organized, and some important definitions are not given. For example, there are two pages for preliminaries and Vision MLP Models before the proposed method. However, there are only two pages for the proposed method. The definition of two important methods Bi-MLP-S and Bi-MLP-M are missing.***
>
> A1.5: Thanks for the suggestions and we will try to improve the presentation of the manuscript. We would like to argue that **1+1/5 pages** for the preliminaries is essential to make this paper self-contained, since the background knowledge of BNNs and vision MLPs is necessary to prevent readers from missing the preposition knowledge and is helpful for readers to understand our proposed method. And **2+1/2 pages** of method is enough to give a clear and non-redundant introduction of the proposed Bi-MLP.
>
> Besides, details of the structures of Bi-MLP-S and Bi-MLP-M are already given in Table 1.
>
> ***Q1.6: The devised new framework is not useful. The core contribution combining special projection and channel projection doesn’t achieve a large increase (less than 0.8%) than simple combination of special projection and channel projection (setting 1 in Table 3).***
>
> A1.6: Rather than simple combination, all the settings in Table 3 still use the proposed MBB Blocks. The only difference between setting 1 and setting 3 is that the elements in MBB Block 1 and 2 are different. We also keep (#S1 + #S2) / (#C1 + #C2) = 2 so that the ability of fusing information of different dimensions is roughly the same as mentioned in Line 267-271. Besides, increasing the top-1 accuracy by 0.8% is significant enough on the challenging ImageNet benchmark.

---

> ### Author Response · Authors · 2022-08-01
> **Rebuttal to Reviewer orUW (Questions & Limitations part)**
>
> **Rebuttal to Questions:**
>
> ***Q2.1: The advantage of ‘Ops’ is benefit from the downsampling layers. Are downsampling layers also added to the SOTA methods [17][2]?***
>
> A2.1: ReActNet uses binary conv3x3 for downsampling [r7]. Thus, replacing the downsampling layer with the proposed downsampling blocks will drastically increase the OPs since our method use full-precision conv1x1. The advantage of OPs comes from the whole architecture rather than the downsampling layers.
>
> ***Q2.2: In the ablation study, the short-cut, BN+RPReLU, different MBB block settings achieve about 0.9%, 0.6%, and 0.8% increase, respectively. The overall framework only achieves 70% (Bi-MLP-S) and 72.2% (Bi-MLP-M) accuracy. The SOTA method (ReActNet) achieve 71.4% accuracy. So, which part brings the fundamental increase compared with SOTA methods?***
>
> A2.2: First of all ,our method achieves 72.7% accuracy for Bi-MLP-M. Note that the SOTA method ReActNet uses CNN model as base architecture, while we use MLP as base architecture. Directly applying the methods in ReActNet to MLP yields only 63.2% accuracy in Table 7. In our method, MBB Blocks and Uni-shortcut are both important for the performance increase, and BN+RPReLU is also used in ReActNet and is used in our method for fair comparison.
>
> Besides, we have 12.1% less OPs compared to the ReActNet model with 71.4% accuracy.
>
> ***Q2.3: What’s the meaning of Bi-MLP-S and Bi-MLP-M?***
>
> A2.3: See Table 1 in the original paper for details. We will include more explanation to avoid any confusion.
>
> ***Q2.4: Does the SOTA method also adopt the same data-augmentation strategy? The 1.3% increase is benefit from the data-augmentation strategy. The author is suggested to add the ablation study on the setting with and without data-augmentation strategy.***
>
> A2.4: We conduct experiments on Bi-MLP-S on ImageNet with strong data-augmentation (cut-mix, mixup and rand-augment) and normal data-augmentation strategy used in ReActNet by training 100 epochs due to the limitation of rebuttal time. The experimental results are shown below.
>
> |Method|Top-1 Acc|
> |-|-|
> |strong data-augmentation|67.5|
> |weak data-augmentation|67.6|
>
> Note that using weak data-augmentation strategy is even better than the strong one, since simple models have less trouble with overfitting and simple data augmentation is enough for a good training result.
>
> **Rebuttal to Limitations:**
>
> ***Q3.1: The improved performance is limited. The proposed method only achieves 1.3% increase.***
>
> A3.1: See A1.2 in the rebuttal of weaknesses part.
>
> ***Q3.2: From Table3, we can see that setting 3 only achieves 0.8% increase than setting 1, which is a simple combination of special projection and channel projection. So I suspect that the main contribution of the combination of special projection and channel projection is not really useful.***
>
> A3.2: The main contribution of MBB Block is to increase the representation ability of BNN with multi-branches rather than the combination of spatial projection and channel projection. The specific elements in MBB Block is flexible. We believe that balancing the representation ability in each dimension is a good way to further increase the performance and Table 3 validate this hypothesis.
>
> ***Q3.3: The proposed method is only validated on one dataset.***
>
> A3.3: See A1.4 in the rebuttal of weaknesses part.
>
> ***Q3.4: The proposed method is the integration of existing techniques.***
>
> A3.4: Note that in Table 7, directly apply different binarization techniques in CNN to MLP yields bad performance. We found that the poor representation ability in binary FC layer is the central problem for MLP and propose a brand new architecture for binarizing MLP.
>
> ***Q3.5: The author only gives the Ops performance with block setting from an experimental viewpoint. The author is suggested to add the theoretical analysis of inference performance with different block settings.***
>
> A3.5: We give the analysis of computational complexity and representation ability in Line 151-160. More detailed theoretical analysis will be added in the final version.
>
> [r1] ReCU: Reviving the Dead Weights in Binary Neural Networks. ICCV 2021.
>
> [r2] Learning Frequency Domain Approximation for Binary Neural Networks. NeurIPS 2021.
>
> [r3] AdaSTE: An Adaptive Straight-Through Estimator to Train Binary Neural Networks. CVPR 2022.
>
> [r4] Improving Accuracy of Binary Neural Networks using Unbalanced Activation Distribution. CVPR 2021.
>
> [r5] Lipschitz Continuity Retained Binary Neural Network. ECCV 2022.
>
> [r6] Equal Bits: Enforcing Equally Distributed Binary Network Weights. AAAI 2022.
>
> [r7] https://github.com/liuzechun/ReActNet/blob/master/mobilenet/2_step2/reactnet.py

---

> ### Author Response · Authors · 2022-08-07
> **Any further questions?**
>
> Dear reviewer orUW:
>
> We sincerely thank you for the review and comments. We have provided corresponding responses and results, which we believe have covered your concerns. We hope to further discuss with you whether or not your concerns have been addressed. Please let us know if you still have any unclear parts of our work.
>
> Best,
> Authors of Paper 7915

---

> ### Comment · Area_Chair_jSMH · 2022-08-09
> **Rebuttal Response**
>
> Dear Reviewer,
>
> could you please indicate that you have considered the authors' rebuttal? (E.g. by replying to the rebuttal or at least by using the "Author Rebuttal Acknowledgement".)
>
> The [reviewer guidelines](https://nips.cc/Conferences/2022/ReviewerGuidelines) ask: "Even if the author response didn’t change your opinion about the paper, please acknowledge that you have read and considered it."
>
> Thank you!

---

### Meta-Review · Area_Chair_jSMH · 2022-08-26

**Recommendation:** Accept
**Confidence:** Less certain

**Metareview:**

Four reviewers provided feedback on this paper. The authors provided a response to the reviews and I appreciate the authors' detailed comments and clarifications, specifically addressing each reviewer's comments/questions. The authors did not upload a revised version of the paper.

After the two discussion periods, three of the four reviewers suggest to accept the paper (with varying scores) while one reviewer (orUW) rated the paper as "borderline reject", so not strongly opposing acceptance. (Also, reviewer orUW chose to not engage in the discussion, nor did they acknowledge the authors' response, so their opinion should carry slightly less weight in the overall decision.) After considering the reviewers' and authors' comments, I believe that the paper should be accepted to NeurIPS.

Weaknesses include:
* The approach is only validated on one dataset.
* Concerns regarding missing related work. (Partially addressed in the response.)
* It would be great to see also concrete runtime or throughput numbers, not only OPs counts.

Strengths include:
* Novel approach to binary networks for image classification. Interesting discussion of binary networks in the context of CNN/MLP approaches.
* New SOTA on ImageNet for binary networks (using fewer OPs).
* Experiments support claims.

I expect and hope that the authors will thoroughly address the reviewers' comments in the camera ready version of the paper.

Minor points (not affecting this decision, but potentially useful to authors when preparing the final revision):
* "sign function is non-differentiable almost everywhere" - it seems to me that the sign-function is actually differentiable almost everywhere, but has a zero gradient, maybe that is what is meant?
* typo: "Dowmsampling" (291)

**Award:**

No

---

### Decision · Program_Chairs · 2022-09-14

Accept